# An Investigation of Takagi-Sugeno Fuzzy Modeling for Spatial Prediction with Sparsely Distributed Geospatial Data

Robert Thomas [1], Usman T. Khan [2], Caterina Valeo [1,*] and Mahta Talebzadeh [1]

1. Department of Mechanical Engineering, University of Victoria, Victoria, BC V8W 2Y2, Canada; thomas.rwc@gmail.com (R.T.); talebzadeh@uvic.ca (M.T.)
2. Department of Civil Engineering, Lassonde School of Engineering, Toronto, ON M3J 1P3, Canada; usman.khan@lassonde.yorku.ca
* Correspondence: valeo@uvic.ca

**Abstract:** Fuzzy set theory has shown potential for reducing uncertainty as a result of data sparsity and also provides advantages for quantifying gradational changes like those of pollutant concentrations through fuzzy clustering based approaches. The ability to lower the sampling frequency and perform laboratory analyses on fewer samples, yet still produce an adequate pollutant distribution map, would reduce the initial cost of new remediation projects. To assess the ability of fuzzy modeling to make spatial predictions using fewer sample points, its predictive ability was compared with the ordinary kriging (OK) and inverse distance weighting (IDW) methods under increasingly sparse data conditions. This research used a Takagi–Sugeno (TS) fuzzy modelling approach with fuzzy c-means (FCM) clustering to make spatial predictions of the lead concentrations in soil. The performance of the TS model was very dependent on the number of outliers in the respective validation set. For modeling under sparse data conditions, the TS fuzzy modeling approach using FCM clustering and constant width Gaussian shaped membership functions did not show any advantages over IDW and OK for the type of data tested. Therefore, it was not possible to speculate on a possible reduction in sampling frequency for delineating the extent of contamination for new remediation projects.

**Keywords:** fuzzy modelling; marine sediment; Takagi–Sugeno; ordinary kriging (OK); inverse distance weighting (IDW); spatial predictions

## 1. Introduction

The release of pollutants into the natural environment has been a problem of global concern since the beginning of the industrial revolution. Highly toxic persistent environmental pollutants often occur in marine harbour sediments as a result of industrial practices around the world and pose a significant risk to human health [1]. The major contributor to exposure of humans to contaminated sediment is through the ingestion of contaminated food as a result of bioaccumulation through the food chain. Chronic exposure to contamination can lead to infertility, birth defects, impaired child development, diabetes, damage to the immune system, disruption of hormonal function, and cancer. Seafood in the most common exposure pathway of sediment contamination to humans; therefore, for the protection of human health, the remediation of these contaminants in aquatic environments is of the upmost importance [2–5].

The first step required for remediation is an accurate assessment of the spatial distribution of contamination in order to ensure the most effective and least costly remediation. Spatially continuous data are required to delineate the boundaries of unsafe levels of contamination and to determine the volume of contaminated material to be removed. However, in aquatic environments, point samples are generally collected on a predetermined grid spacing, and the contaminant concentrations are spatially interpolated to provide a continuous surface that may introduce uncertainty. Spatial interpolation methods are traditionally grouped into deterministic and geostatistical methods. The most commonly

used deterministic method is inverse distance weighting (IDW), which uses a function that estimates values at un-sampled points through the linear combination of known sample values weighted by the distance between them [6,7]. IDW is a simple method requiring minimal modeler input [8]; however, because it is based solely on distance, it often performs very poorly with sparsely distributed geospatial data. The most commonly used geostatistical method for spatial interpolation is kriging, which employs a semi-variogram that plots the semi-variance between points against the distance between them [6,9]. From the semi-variogram, it is possible to determine the range of spatial dependence of sampled points that can be used to determine a value at an unknown location [10]. However, the accurate estimation of a semi-variogram is complicated, computationally expensive, and can introduce modeler bias. Furthermore, kriging has been shown to have a significant smoothing affect, where areas of high pollutant concentration could be missed. This is not ideal as an underestimation of the pollutant concentrations could lead to an increased risk to human health. Both deterministic and geostatistical methods have one major commonality, that is, the greater the sample density, the greater the accuracy of the spatial interpolation [11–13]. However, the sampling cost of sediment in marine environments and the analytical assessment cost for dioxins are extremely high. Therefore, obtaining an adequate number of samples to achieve an acceptable resolution during interpolation may not be possible, and this high cost may be prohibitive to remediation projects.

There is a separate family of data-driven predictive methods that utilize fuzzy set theory [14], which have been proven to be a suitable method for the prediction of sparse non-linear data and have been used for many applications of spatial estimation in geoscience [11,15–17]. Fuzzy set theory [14] provides a convenient way of describing the degree of belonging (membership $\mu$) of an element to a set, between 0 (no belonging) and 1 (complete belonging). For example, let U be an ordinary set with elements $\{x_1, x_2, \ldots, x_n\}$ and $\widetilde{A}$ be a fuzzy subset of U, in which the elements $x_i$ have degrees of membership (belonging to $\widetilde{A}$) given by a membership function $\mu_{\widetilde{A}}(x_i) = \alpha$, which dictates that an element $x_i$ has a degree of membership $\alpha$ to fuzzy set $\widetilde{A}$, where $0 \leq \alpha \leq 1$ (see Figure 1).

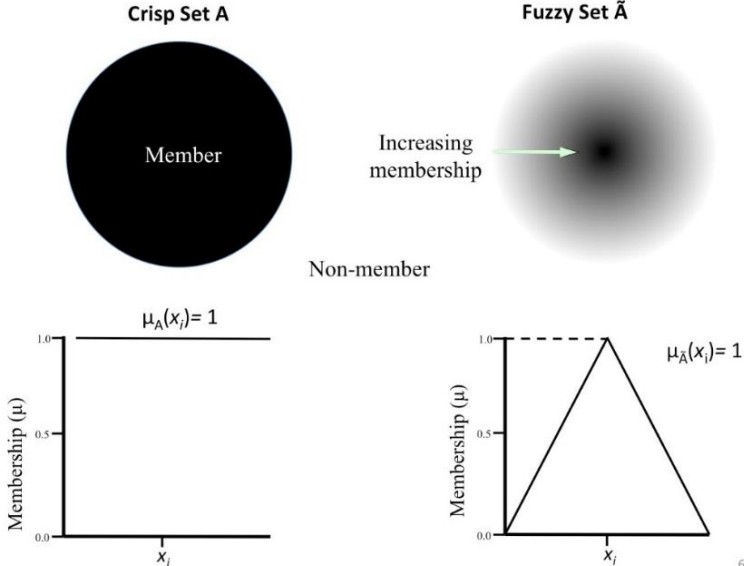

**Figure 1.** Example of a (**left side**) crisp and (**right side**) fuzzy set, and their respective membership functions.

*The Takagi-Sugeno Method*

System modeling techniques that employ fuzzy set theory are commonly referred to as fuzzy modeling [15,18,19]. One of the most commonly used fuzzy modeling techniques for spatial estimation is the Takagi–Sugeno (TS) method [15,18,20]. TS fuzzy modeling breaks

down the input data space into a number of fuzzy regions and creates a linear function for each region. It is advantageous for spatial modeling because of its transparency and interpretability [21]. The degree of belonging that an un-sampled point has to each region in the data is used to predict a value at that location. The partitioning of the input space into fuzzy regions is achieved though fuzzy clustering, which is the foundation for the fuzzy spatial modeling using the TS method. Fuzzy clustering differs from traditional crisp data clustering in that in fuzzy methods each element (sample point) can have a degree of membership to multiple clusters within the data. Each cluster is defined by a cluster center, which has a value calculated from the membership-weighted average of the members of that cluster [22,23]. For spatial modeling, data are clustered in the three-dimensional product space defined by the Cartesian map coordinates $(x, y)$ and the magnitude of a pollutant concentration $(p)$ (see Figure 2).

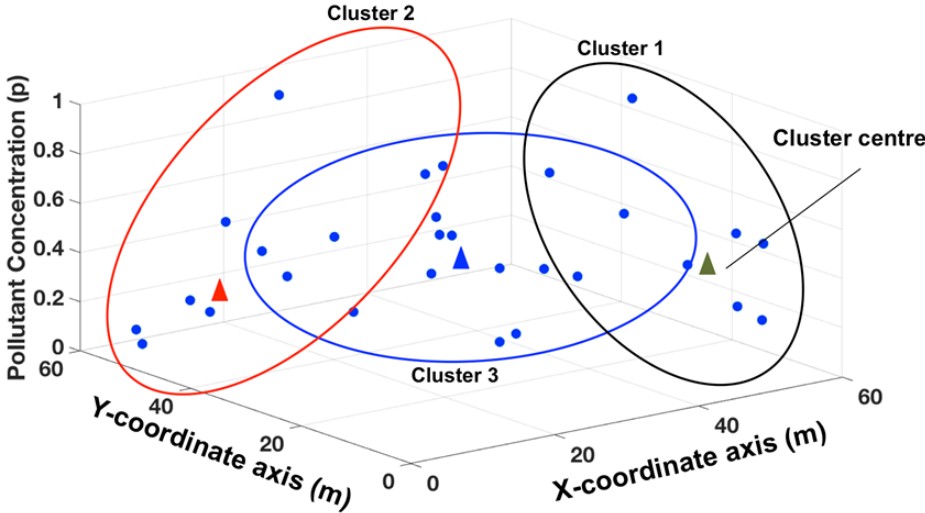

**Figure 2.** Theoretical visualization of fuzzy clustering in the three-dimensional product space.

After clustering, each data point has membership to one or more cluster centers, depending on the pollutant concentration. The clusters are then partitioned onto the $x$ and $y$ Cartesian product space and a membership function is generated for the $x$ and $y$ coordinate axis of each cluster. The membership functions from each cluster are then used to determine the degree of membership an un-sampled location has to the different clusters within the data (see Figure 3).

Ultimately, the combination of these degrees of membership is used in solving a pollutant concentration at an unknown location. Once the data have been clustered, they are subjected to a rule-based fuzzy inference system (FIS) that makes inferences about un-sampled geographic locations based on their membership to the clusters within the data. A single rule is introduced for each cluster using conditional "IF-THEN" statements. FIS uses input variables referred to as antecedents for each rule; in the TS fuzzy model, the antecedents are the fuzzy sets generated from the clustered data. The antecedents are subjected to IF-THEN statements to produce a weighting for the prediction from each rule based on membership to that rule (cluster). The output of each rule is referred to as a rule consequent. When a rule in the FIS is executed, if the antecedent is unaffected by the IF-THEN condition, that rule is skipped and the next rule is executed. If the IF-THEN condition produces a consequent, then that rule is deemed to have fired or been executed.

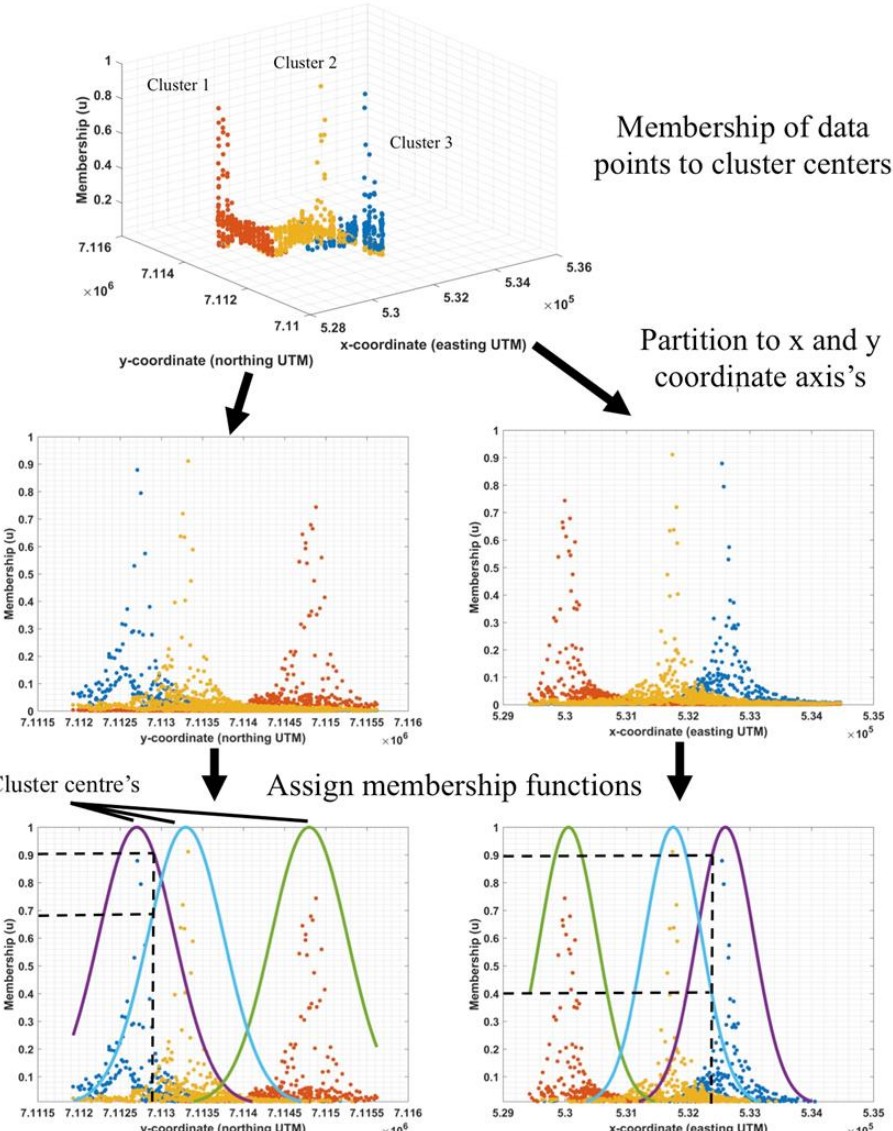

**Figure 3.** Example of partitioning membership to cluster centers to the map coordinate axes and applying a membership function (represented by solid lines) for three fuzzy clusters to determine membership based on geographic location.

Each rule in the TS fuzzy model uses *x* and *y* Cartesian coordinates as inputs to solve the consequent (output variable) for each rule in the form of a linear equation to produce a crisp value [23]. The form of the general first order TS model is shown in Equation (1).

$$R_i = if \ x_1 \ is \ A_{i1} \ and \ x_2 \ is \ A_{in} \ then \ p_i = a_{i1}x_1 + a_{i2}x_2 + b_i$$
$$i = 1, \dots, K \tag{1}$$

where $R_i$ is the *i*th rule; $x_1$ and $x_2$ are the antecedent variables (Cartesian coordinates *x*, *y*); $A_{i1}$ and $A_{i2}$ are fuzzy sets for the *i*th rule and the respective coordinate axes; $p_i$ is the *i*th rule output; *K* is the number of rules (clusters); and $a_{i1}, a_{i2},$ and $b_i$ are the unknown model parameters that must be solved. This is accomplished by least squares regression using the data in each cluster. The model output for a given input is obtained through the aggregation of all utilized rule consequents weighted by their membership to the respective rules [15,19,20]. The TS method's simplistic nature and interpretability make it advantageous compared to other FIS [24].

Applications using TS fuzzy modeling for spatial estimation have shown its predictive capacity to outperform kriging [19,25]. The advantage of TS modeling is that it allows a complex data surface to be broken down into more easily modeled individual fuzzy surfaces. Clustering and rule-based methods are then used to estimate a value at unknown locations by solving the intersection of the contributing surfaces based on the degree of belonging an unknown location has to each surface.

Fuzzy set theory has been cited as a useful tool for modeling under sparse data conditions [26]. Fuzzy modeling can take advantage of this additional information to produce an adequate contaminant distribution map for remediation planning using fewer point samples. Lowering the number of samples required would reduce the initial cost of new remediation projects and may lead to more remediation projects being undertaken in the future, eventually leading to a cleaner environment and to lower anthropogenic health risks [27].

This research employs a spatially distributed marine, soil geochemical dataset to compare the predictive abilities of the TS fuzzy model, IDW method, and ordinary kriging method (OK) using a sparse dataset. The geochemical signatures present in the data are the result of naturally occurring mineralization and are not related to anthropogenic sources. The data are useful, because the indicator geochemical signatures associated with the mineralization are analogous with several pollutants of concern (POC) in harbour settings. The spatial distribution of the contaminates in the data are very similar to that from anthropogenic sources, as both generally originate from point sources and spread out into the surrounding environment.

## 2. Materials and Data

The dataset used in this research is comprised of 1535 spatially distributed sample points (illustrated in Figure 4), each consisting of a 51 element spectral array with notable POCs of arsenic, mercury, and lead. Although no two spatial datasets would have identical statistical distributions, the challenges faced during spatial interpolation are similar regardless of the data type. Therefore, the use of this data still provides an adequate initial test of the relative predicative ability of the TS fuzzy model under increasingly sparse spatial data density.

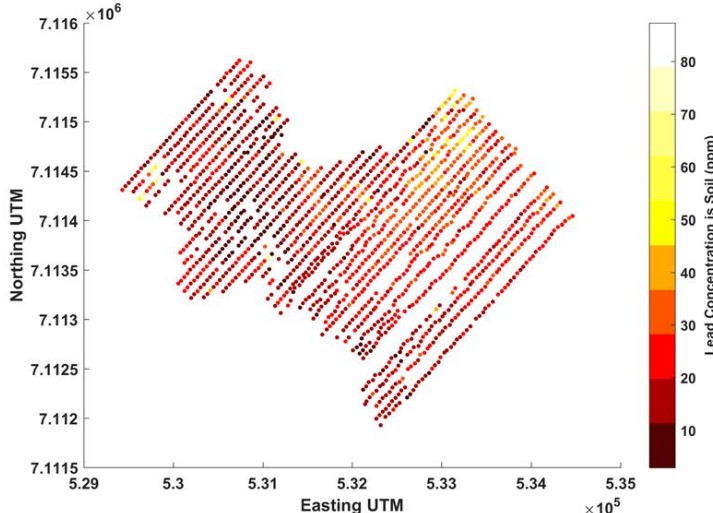

**Figure 4.** Spatial expression of the data used for this research, Datum: NAD83, UTM Zone 8, Eastern Yukon Territory.

The 1535 soil samples were collected on an approximately 50 m by 50 m grid spacing at depths ranging from 0.1–1.2 m below the ground surface. The samples had compositions ranging from well to poorly developed soils to glacial sediment. The sample area was

approximately 5000 m × 4000 m in size and was in the Yukon Territory, Canada. The samples were collected between 17 August 2015 to 22 August 2015, and between 16 June 2016 to 25 June 2016 by a sampling team from Archer, Cathro, and Associates Limited under contract for ATAC Resources Limited. ALS Minerals in Vancouver BC analyzed the samples using metallurgical assay with inductively coupled plasma and atomic emission spectrometry. The spatial coordinates of each sample were recorded in the spatial datum NAD83 UTM Zone 8. Each data point was characterized using an easting and northing UTM coordinate in meters. Because of its relevance to human health, its high rank as a POC in urban settings and its availability in the soil data, lead was selected to test the efficacy of TS fuzzy modeling compared with OK and IDW in this work.

As OK and IDW are well-developed tools, this research employed the GIS platform ArcMap 10.4.1 (ESRI, 2011) and used the Geostatistical Analyst Package to perform the OK and IDW methods using optimized parameters. Using the spatial predictions from ArcMap, model validation was performed using MATLAB R2017a (MathWorks, 2017). The fuzzy model was also developed and validated, and its results were compared with OK and IDW using MATLAB.

In order to assess the predictive ability of the TS fuzzy, kriging, and IDW methods, the sample data were split into training and validation sets, where the training data were used to predict the known pollutant concentrations at the locations in the validation set. To assess the predictive ability of the models under increasingly sparse spatial data conditions that simulated fewer samples, the amount of training data used were incrementally reduced from 75%, to 66%, to 50%, to 33%, to 25%, to 12.5% and labelled as sets A–F, respectively. At each increment, the data not used for training were utilized for validation.

To determine the optimal number of clusters, the fuzziness parameter ($m$) for the FCM clustering and the membership function width ($\sigma$) for the FIS, a pseudo-optimization was performed. This was accomplished by iteratively increasing the number of clusters and at each iteration varying $m$ through a reasonable range, and at each iteration of $m$ varying $\sigma$ through a reasonable range. The model performance results from all combinations were then assessed and the combination of $m$ and $\sigma$ that yielded the most accurate prediction for the respective validation sets were retained as optimal.

The key things that determined the competency of a spatial interpolator were its predictive ability, its ability to handle data of different types and variance, and the smoothness or abruptness of the surface generated [6]. The performance metrics used to assess the model performance of TS fuzzy, OK, and IDW modeling were the coefficient of determination ($R^2$), root mean squared error (RMSE), and the mean absolute error (MAE). $R^2$ is commonly used and provides a metric by which to assess the variance of the model prediction. A higher $R^2$ value indicates that a higher proportion of the total variation of the prediction is explained by the model [19]. An $R^2$ value equal to 1 would indicate a perfect prediction. Each model result for MAE, $R^2$, and RMSE received a score between 1 and 10. The mean score then determined the overall performance of that model result, with 10 being the best and 1 being the worst. The ranges for the scoring system were chosen arbitrarily based on the performance ranges observed from each metric. This approach aimed to produce enough separation between scores to draw realistic inferences on the differences in performance.

## 3. Results

When determining the optimal number of clusters, it was observed that the performance of the of TS fuzzy model would reach an initial peak or plateau and subsequent model runs using a higher number of clusters would not produce any further increase in score. In the interests of maintaining minimal complexity within the model and reducing the risk of over clustering, the lowest number of clusters that yielded the highest model performance was retained as optimal. In cases where more than one TS fuzzy model iteration with the optimal number of clusters but different parameters produced the same mean score, the result with the lowest mean absolute error (MAE) was deemed optimal.

The MAE was selected because of its prevalence in assessing the accuracy of spatial predictions [28]. Figure 5 displays the results from the selection of the optimal number of clusters for data reduction increments A and B for subset 1. In each case, the highest mean score from each number of clusters tested is displayed. Therefore, for each cluster, the results displayed are the optimal performance with the optimal $m$ and $\sigma$ for that number of clusters.

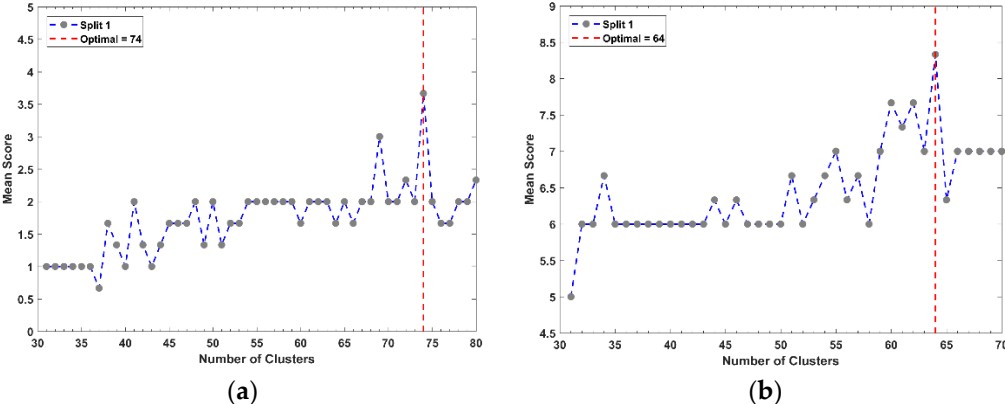

**Figure 5.** Example results from the determination of the optimal number of clusters for data reduction for (**a**) increment A and (**b**) increment B, for subset 1.

In all cases, a clear initial peak was reached and dictated the optimal number of clusters for that particular training set. After identifying the number of clusters that yielded the highest model performance, the combination of fuzziness ($m$) and membership function width ($\sigma$) that contributed to the highest mean score were identified. Figure 6 displays an example of the results from determining the optimal $m$ value. In each case, the highest mean score for each $m$ value tested with the optimal number of clusters is displayed.

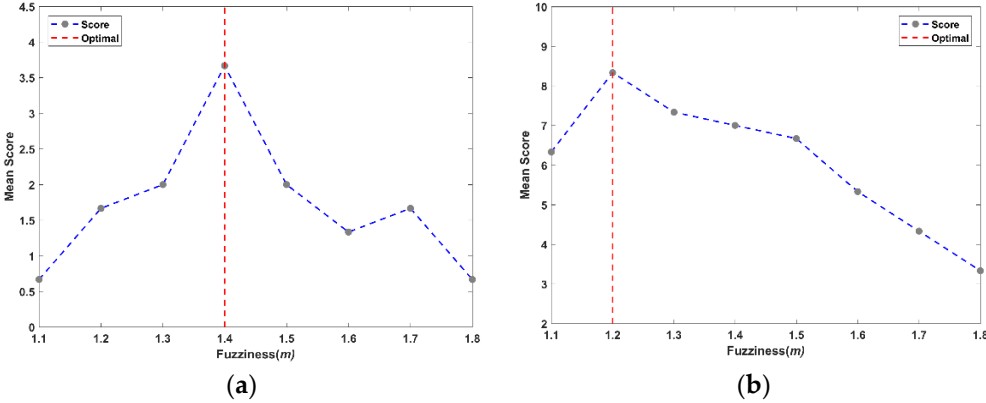

**Figure 6.** Example results from the determination of $m$ using the optimal number of clusters for data reduction for (**a**) increment A and (**b**) increment B.

In all instances, a single $m$ value produced the highest model performance for each increment of the data reduction and the $m$ value had a visibly large effect on the performance of the TS fuzzy model. The optimal $\sigma$, which produced the highest model performance, was determined to occur over a range and was not overly sensitive. In general, a range of 80 m to 150 m produced similar mean scores. Figure 7 displays a 2D visual representation of the clustered data, the cluster centres, and the spatial distribution of membership to the cluster centres. The concentration of lead at a validation point was solved using its membership to each of the clusters in the data.

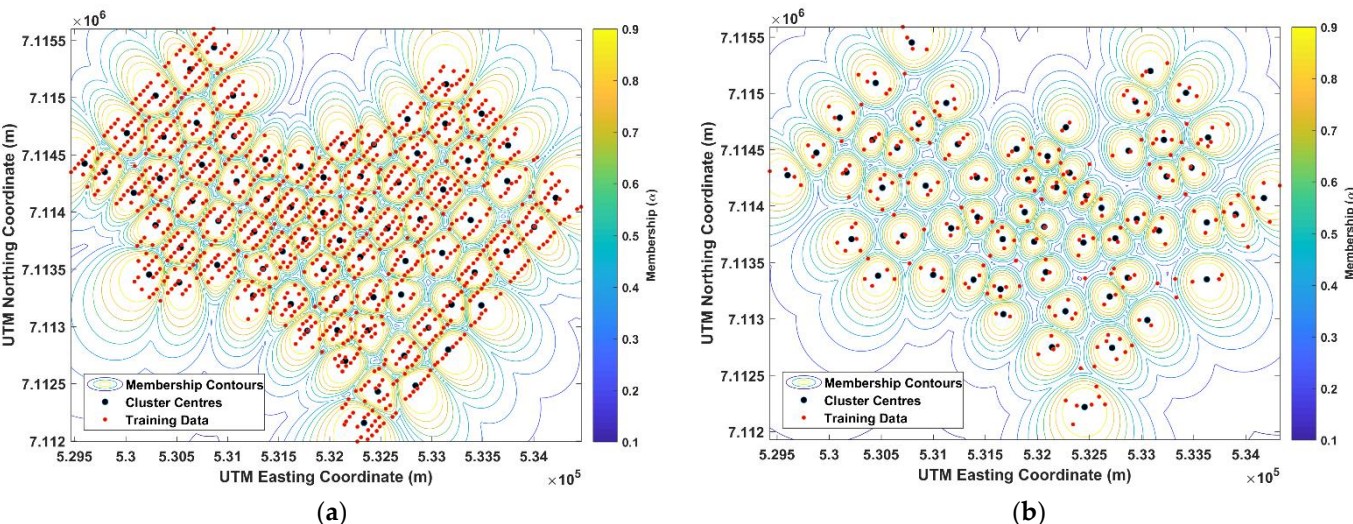

**Figure 7.** (**a**) Visual representation of clustered data for training increment A and (**b**) training increment F.

Finally, using the optimal model parameters, the fuzzy model was trained using each increment's training data (A–F); then, the lead concentrations at the spatial locations of each increment's validation sets were solved using the model. The predictions were then compared with the actual values and the performance metrics applied. This same methodology was also applied to the OK and IDW methods, such that the performance of the three models could be compared. Figure 8 displays the performance results for each model at all of the data reduction increments.

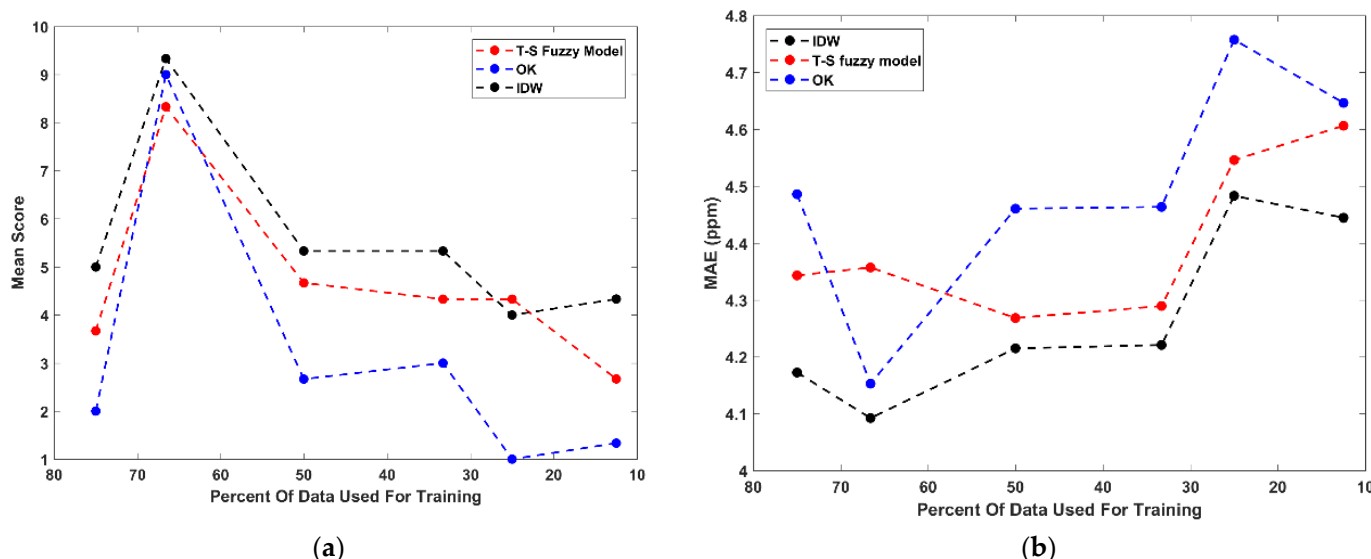

**Figure 8.** *Cont.*

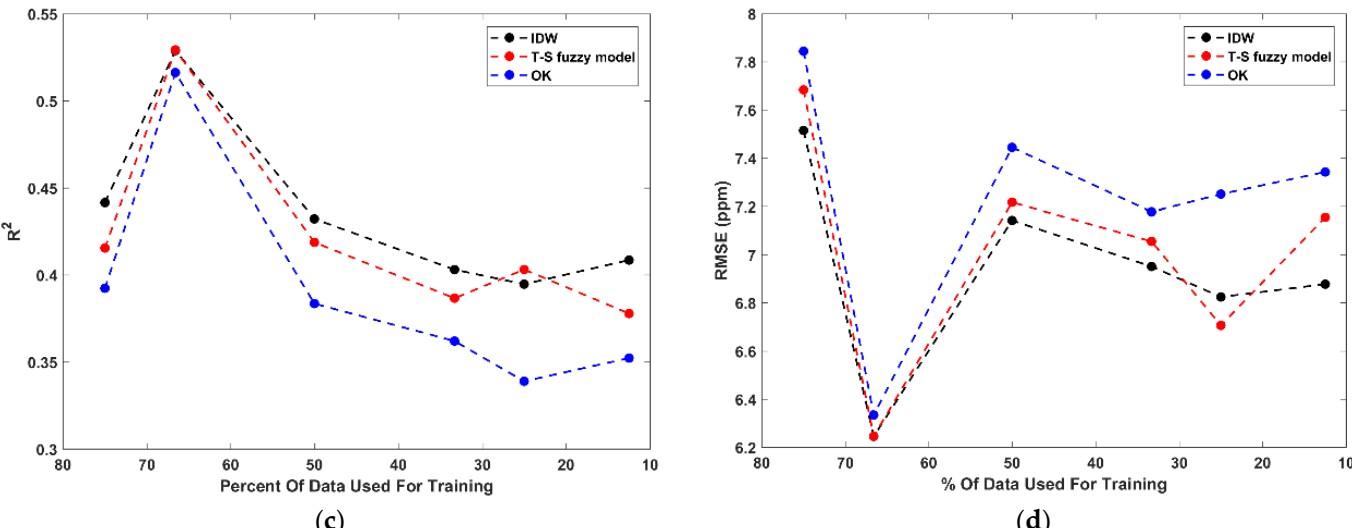

**Figure 8.** Ordinary kriging (OK), inverse distance weighting (IDW), and the Takagi–Sugeno (TS) fuzzy model from each increment of data reduction performance metrics (**a**) mean score, (**b**) mean absolute error (MAE), (**c**) coefficient of determination (R$^2$), and (**d**) root mean squared error (RMSE).

## 4. Discussion

In general, the three models tested here performed very similarly. For all three methods, data reduction increment B produced the highest performance scores. This is because this increment's training set has a much higher range, variance, and kurtosis than the respective validation set, making for simpler predictions closer to the mean. The discrepancies identified between the training and validation sets appear to have a large effect on the performance of the models—specifically, between increment A and B, where, for increment A, the kurtosis and range are higher in the validation set and for increment B, the kurtosis and range are higher in the training set. The effect of statistical differences between the training and validation sets does appear to be significant. However, in a real-world situation, if fewer samples are collected and analyzed, the results could have a lower range and kurtosis; so, for the purposes of simulating spatial data sparsity, the training and validation sets used here are useful because they illuminate weaknesses in the models tested. This proves that if the samples collected fail to capture the overall variance of a system, their spatial predictions will be poor, which Li and Heap (2011) found when reviewing 18 studies using different spatial interpolators.

Throughout the data reduction increments, on average, all of the methods' performances decline, which is the desired result of the evaluation. For increments A–F, IDW consistently outperformed the other methods, and the TS fuzzy model outperformed OK, but only by a small margin. To further quantify these observations, the performance of the individual metrics from the optimal TS fuzzy model results and for the OK and IDW methods from increments A–F are also plotted against the amount of training data used in each increment (shown in Figure 8). The differences between the three methods are so subtle that it is difficult to draw any significant conclusions from the results. As is consistent with the mean scores, based on these performance metrics, IDW appears to produce the best results, with the TS fuzzy model outperforming OK by a small amount. For the lead predictions made in this analysis, OK, IDW, and the TS fuzzy model all appear to have a significant smoothing effect at all data reduction increments, which agrees with previous research [12,29]. Previous research comparing IDW and OK found that in general, the two methods perform very similarly, and that IDW often produces a lower error, and this research agrees with that finding [29–31]. Furthermore, previous research has also shown that TS fuzzy modeling has the ability to outperform kriging [25,32,33] as it has in this study, albeit by a small amount. Although not clear in this research, the TS fuzzy model

did produce a higher mean score than OK for data reduction increments C–F. Although the three models are not using identical distributions in their predictions of unknown points, it is possible that the similarity in the results is simply a function of the inherent similarities within the models themselves. However, further research would be required in order to quantify this.

As fuzzy clustering is at the core of the TS fuzzy modeling approach, it can be surmised that the performance of the method may be limited by the quality of the fuzzy clusters or by their spherical shape, which agrees with findings from the literature [15]. Therefore, further investigation into other clustering methods is warranted. Additionally, the data used in this research have a negatively skewed distribution. A factor that was not considered was normalizing the distribution of the data prior to the analysis, which would impact the performance of the models. However, for testing the relative performance of the TS fuzzy model against IDW and OK for spatial prediction using incrementally less spatially distributed data, the results do show that the TS fuzzy model in this research is not a superior prediction method.

A useful confirmation from this research is that fuzzy set theory is highly flexible [14] and thus, has many other applications including the prediction of time dependent variables [26]. Fuzzy-based methods lend a flexibility to environmental modelling and assessment that crisp methods do not [34,35]. It can be incorporated into other types of modelling schemes such as those that consider connectivity in earth system processes, for example. Connectivity is a subject that describes the degree to which water flow and sediment are related or connected [36,37], and can be used to acquire enough accurate data of past and current conditions to assess the role of landscape processes. When contaminants are costly to analyze, as is the case for many emerging contaminants of concern in coastal regions, Keesstra et al. [36] showed that a connectivity modelling scheme can be used to help define a more cost-effective monitoring and measurement plan.

## 5. Conclusions

A TS fuzzy model using the fuzzy c-means clustering algorithm was used to predict lead in soil concentrations under increasingly sparse spatial data conditions. The results from the TS fuzzy model were compared with that of OK and IDW using the same training and validation datasets. The ability of the three methods to predict outlier points within the respective validation sets appeared to be very similar. As with all other performance metrics, the performance of the three methods was difficult to separate and the similarity in model performance may be related to the models themselves. Both IDW and OK generate weights for the surrounding points based on a similar model shape, while the TS fuzzy model utilizes Gaussian shaped membership functions to determine the weighting for each of the cluster's predictions. This research showed that the TS fuzzy model used here did not outperform OK and IDW for the spatial predictions of lead in soil under increasingly sparse data conditions. Future research should consider different types of data, normalizing data prior to fuzzy modeling, other clustering methods, and further optimization of the model parameters.

**Author Contributions:** Conceptualization, R.T., U.T.K. and C.V.; methodology, R.T., U.T.K. and C.V.; validation, R.T.; formal analysis, R.T.; investigation, R.T.; resources, C.V.; data curation, R.T.; writing—original draft preparation, R.T. and M.T.; writing—review and editing, R.T., U.T.K., C.V. and M.T.; supervision, C.V. and U.T.K.; project administration, C.V. All authors have read and agreed to the published version of the manuscript.

**Funding:** This research received no external funding.

**Acknowledgments:** The authors would like to thank the reviewers for their insightful comments in their reviews of this work. The authors would also like to thank the Department of Mechanical Engineering at the University of Victoria for access to software used in this work.

**Conflicts of Interest:** The authors declare no conflict of interest.

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
