# Peer review of "An Investigation of Takagi-Sugeno Fuzzy Modeling for Spatial Prediction with Sparsely Distributed Geospatial Data"

_environments, doi:10.3390/environments8060050_

Round 1

Reviewer 1 Report

The utility of fuzzy set based methods to generate spatial distributions form sparse data is examined and compared with results from kriging and inverse distance weighting methods. This is done using a dense set of pollutant data from a marine location, sub-sampled to give sparse data and model generated distributions tested against the remaining data.  It is concluded that the fuzzy model is no better than the other techniques, and that the inverse distance method is marginally better than the other methods on the dataset used.  While this is a somewhat negative result, it is important that such results are published to avoid biases in the literature.

The work is well presented and carefully conducted, with no significant changes needed. You could perhaps show a couple of plots of predicted distribution at the sample locations (i.e. similar to Fig 4, but predictions, plus some way of highlighting the "training points") to give the reader a visual impression of the predicted fields.  Some minor corrections by line number below.

98 membership of one or more

126 parameters which must be solved. This is accomplished..

241 each increment's

242 increment's

257 increment's

Author Response

Reviewer 1

We would like to thank this Reviewer for taking the time to review this paper and provide their insightful comments. We are grateful for their support of the work. Reviewer comments shown in italics and response shown in normal text.

  1. The utility of fuzzy set based methods to generate spatial distributions form sparse data is examined and compared with results from kriging and inverse distance weighting methods. This is done using a dense set of pollutant data from a marine location, sub-sampled to give sparse data and model generated distributions tested against the remaining data. It is concluded that the fuzzy model is no better than the other techniques, and that the inverse distance method is marginally better than the other methods on the dataset used.  While this is a somewhat negative result, it is important that such results are published to avoid biases in the literature.

We agree wholeheartedly that negative results are important in the literature and all too often ignored or rejected. This is a strong need we wish to support in this work and we are grateful to the Reviewer for this insightful comment.

  1. The work is well presented and carefully conducted, with no significant changes needed. You could perhaps show a couple of plots of predicted distribution at the sample locations (i.e. similar to Fig 4, but predictions, plus some way of highlighting the "training points") to give the reader a visual impression of the predicted fields. Some minor corrections by line number below.

Response: We have now added Figure 7 after line 244 to provide a visualization of the clusters and training points as requested by the Reviewer. 

  1. 98 membership of one or more

Response: The statement currently reads “membership to one or more” is actually correct because the sentence is referencing a data point having membership to one or more clusters. We feel that “membership of one or more” may alter the meaning beyond that which is intended, hence, no change has been made here.

All of these suggested changes for lines 126, 241, 242, 257 have all been made.

  1. 126 parameters which must be solved. This is accomplished.
  2. 241 each increment's
  3. 242 increment's
  4. 257 increment's

Reviewer 2 Report

Dear 

The paper does not have a proper discussion

The quality of the paper is right but I reject the paper due to the lack of a proper discussion (this is what makes an article or scientific paper different from a report)

Please, show which are the implications and contributions of your research in the discussion part

The conclusion section is far too long

The literature is not referenced correctly (see format)

Author Response

Reviewer 2

We would like to thank the Reviewer for taking the time to review this paper and provide their frank comments. They have greatly improved the paper. Reviewer comments shown in italics and response shown in normal text.

  1. The paper does not have a proper discussion. The quality of the paper is right, but I reject the paper due to the lack of a proper discussion (this is what makes an article or scientific paper different from a report)
  2. The discussion should compare with other authors and show the implications of your findings.
  3. You can discuss your contribution of your paper to the connectivity of the flows. See here the work of some colleagues.

Comments #1, 2 and 3 are all pointing to the Discussion section and we would like to thank the Reviewer for bringing the shortfalls in this section to our attention. In addition, the Reviewer kindly provided the references to three journal papers, which we have reviewed, and added two of these to the discussion section and are now in the References section. We have also added seven more references on top of this to place this paper in the context of the literature. Please note that we have intentionally kept the discussion short in keeping with the fact that this is a communication and not a full article paper. Thus, to keep balance in the proportion of the documentation, we have tried to keep the Discussion section to around one page. The following are new text added to the Discussion section:

“Specifically proving that if the samples collected fail to capture the overall variance of a system, their spatial predictions will be poor, which Li and Heap (2011) found in reviewing 18 studies using different spatial interpolators. “

“For the lead predictions made in this analysis OK, IDW, and the T-S fuzzy model all appear to have a significant smoothing effect at all data reduction increments, which agrees with previous research [29,12]. Previous research comparing IDW and OK have found that in general, the two methods perform very similarly and that IDW often produces a lower error; and this research agrees with that finding [29,30,31]. Further, previous research has also shown T-S Fuzzy Modeling has the ability to outperform kriging [25,32,33]. Although not clear in this research, the T-S fuzzy model did produce a higher mean score than OK for data reduction increments C-F. Although the three models are not using identical distributions in their predictions of unknown points, it’s possible that the similarity in the results is simply a function of the similarities within the models themselves. However, further research would be required to quantify this.”

“A useful confirmation from this research is that fuzzy set theory is highly flexible [14] and therefore, has many other applications including prediction of time dependent variables [26].  Fuzzy based methods lend a flexibility to environmental modelling and assessment that crisp methods do not [34, 35]. It can be incorporated into other types of modelling schemes such as those that consider connectivity in earth system processes, for example. Connectivity is a subject that describes the degree to which water flow and sediment are related or connected [36, 37] and can be used to acquire enough accurate data of past and current conditions to assess the role of landscape processes. While the contaminated sediments modelled in this work were effectively deposited through urban disposal operations unrelated to natural pathways, Keesstra et al, [36] shows that connectivity modelling scheme can be used to help define a monitoring and measurement plan for contaminated sediments that are costly to analyze.”

  1. The conclusion section is far too long

Thank you for pointing this out. We have edited the Conclusions section to create a succinct, single paragraph that states what was done, what the outcomes were, and what future directions should be in a few simple sentences. Please see the revised document for this final paragraph with track changes in the cover letter.

  1. The literature is not referenced correctly (see format)

Thank you for pointing out the poor formatting. We have corrected the entire references section to comply with MDPI formatting.

Round 2

Reviewer 2 Report

The paper is now ready for publication